# Effects of Blood Flow Restricted Exercise on Electromechanical Delay and Time to Peak Force after Task Failure: A Randomized Crossover Trial

**Mikhail Santos Cerqueira** [1,*] **, Maria Lira** [2] **, Raiff Simplicio da Silva** [2] **, Marco Machado** [3,4] **and Rafael Pereira** [1]

[1] Research Group in Neuromuscular Physiology, Department of Biological Sciences, Universidade Estadual do Sudoeste da Bahia (UESB), Jequie 45210-506, BA, Brazil
[2] Department of Physical Therapy, Federal University of Rio Grande do Norte (UFRN), Natal 59078-970, RN, Brazil
[3] Laboratory of Physiology and Biokinetic, Faculty of Biological Sciences and Health, Universidade Iguaçu (UNIG) Campus V, Itaperuna 28300-000, RJ, Brazil
[4] Laboratory of Human Movement Studies, University Foundation of Itaperuna, Itaperuna 28300-000, RJ, Brazil
* Correspondence: mikalsantosc@hotmail.com

**Abstract:** Introduction: Electromechanical delay (ED) and time to peak force (TPF) could be used to investigate the central or peripheral sources of performance decline in fatiguing tasks. Exercise with partial blood flow restriction (BFR) has been shown to induce fatigue, but the repercussions of exercise with partial BFR on ED and TPF are unclear. The present study aimed to compare the ED and TPF after an intermittent isometric task until failure with BFR and free blood flow (FBF). Methods: In this crossover randomized clinical trial, 15 healthy and physically active men volunteered in this study. Volunteers performed two intermittent isometric handgrip exercise (IIHE) to failure (72 h apart), combined with either BFR or FBF. Maximum voluntary isometric force (MVIF) concomitant with the electromyographic activity of the wrist and finger flexor muscles were assessed before (PRE) and one minute after (POST) the task failure. Within (PRE vs. POST) and between comparisons (eFBF vs. eBFR) of peak force, time to peak force, rate of force development (RFDpeak) and ED were carried out. Results: No significant between-intervention differences were identified pre- or post-exercise. Peak force and RFDpeak reduced significantly after both blood flow conditions ($p < 0.05$), but without between-condition difference. TPF was statistically higher after exercise only in the FBF intervention ($p < 0.05$). None of the interventions induced a significant change in the ED after IIHE. Conclusion: ED and TPF were similar after BFR and FBF, indicating both conditions induce similar acute performance impairments after IIHE, which seems not to be caused by local (i.e., muscular) factors, but probably by central (i.e., neural) factors.

**Keywords:** muscle fatigue; blood flow restriction; electromyography; muscle performance

## 1. Introduction

Exercise with blood flow restriction (eBFR) consists of low-load exercises (<50% of maximum load) performed with reduced arterial flow and abolished venous return in working muscles [1–3]. Such blood flow restriction (BFR) induces a local (i.e., muscular) hypoxic environment that stimulates strength and muscle mass gains [4,5]. Although the physiological mechanisms are still not well understood, it is proposed that there is high local metabolic stress during eBFR, with greater phosphocreatine (PCr) depletion, accumulation of inorganic phosphate (Pi), protons and lactate, as well as a pH reduction [5–7]. Thus, intra and extracellular disorders induced by hypoxia may anticipate the development of muscle fatigue and cause acute and reversible muscle contractile impairment during eBFR [7–12].

Muscle fatigue has been widely studied through force–time curve and surface electromyography (EMG) signal analysis [13–17]. Electromechanical delay (ED) and time

to peak force (TPF) are two possible parameters to extract from this combined analysis (i.e., time–force curve and EMG) [18,19]. The ED corresponds to the time lag between the beginning of muscle electrical activity and the beginning of muscle force production [20]. The ED is a measure of the total duration of the events from the motor unit action potentials traveling along the sarcolemma to the force transmission along the series of elastic components, initiated by cross-bridge formation (excitation–contraction coupling) [21–23]. The TPF corresponds to the elapsed time between the beginning of force production and the maximum force obtained during a resisted muscle contraction, which allows for evaluation of the functional ability to develop muscle force rapidly [24–26].

Since the strength and hypertrophy gains associated with exercise are influenced by exercise-induced fatigue characteristics (i.e., central or peripheral fatigue), knowing these characteristics is relevant in order to properly manipulate parameters of intensity, volume, and recovery time during training programs. Using the rate of force development (RFD) analysis from force–time curves, a previous study found neural and mechanical function impairments after eBFR [27]. However, the predominance of these neural or mechanical impairments in fatigue development after eBFR is not well established. Thus, the integrated analysis of electromechanical delay (ED) from electromyographic signals and time to peak force (TPF), as well as the rate of force development (RFD) from the isometric force–time curve, can add information regarding factors that contribute to eBFR-induced neuromuscular fatigue. Thus, the present study aimed to evaluate and compare ED, RFD, and TPF after an IIHE until muscle failure with BFR and with free blood flow (eFBF).

## 2. Results

The time to failure during IIHE was longer in eFBF than eBFR (eFBF $510 \pm 240$ s vs. eBFR $390 \pm 210$ s; $p < 0.05$). Peak force and contractile RFDpeak reduced significantly after eBFR (Peak force: mean difference (MD) = 180.7 N; 95% confidence interval (CI) = 151.5 to 209.9 N; RFD: MD = 198.6 N·m·s$^{-1}$; 95% CI = 79.2 to 318.1 N·m·s$^{-1}$) or eFBF (mean difference (MD) = 162.0 N; 95% confidence interval = 127.7 to 196.3 N; RFD: MD = 214.9 N·m·s$^{-1}$; 95% CI = 99.8 to 329.9 N·m·s$^{-1}$), but without significant between-condition differences ($p > 0.05$). TPF increased significantly only after eFBF (TPF: MD = $-0.07$ s; 95% CI = $-0.11$ to $-0.03$ s), but no difference was identified between the studied conditions ($p > 0.05$). No difference was identified in within ($p > 0.05$) and between conditions ($p > 0.05$) for ED. Figures 1–4 present the comparisons.

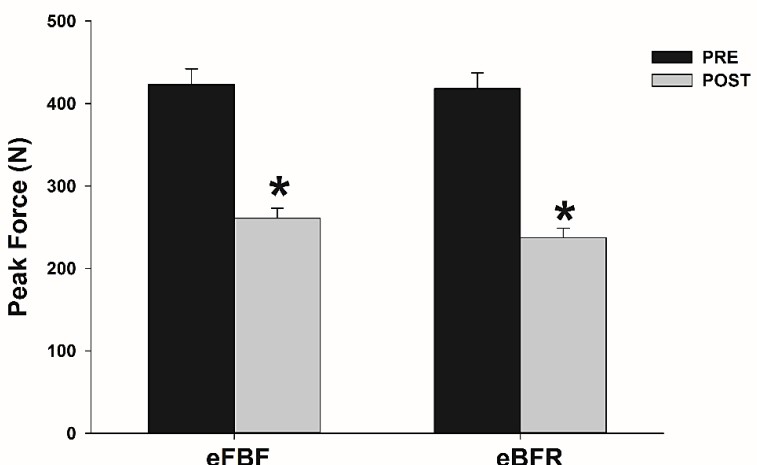

**Figure 1.** Mean $\pm$ SE of peak force before (PRE) and immediately after (POST) the isometric intermittent handgrip exercise until failure carried out with free blood flow (eFBF) or blood flow restriction (eBFR). (*) Significantly different from PRE at the same blood flow condition.

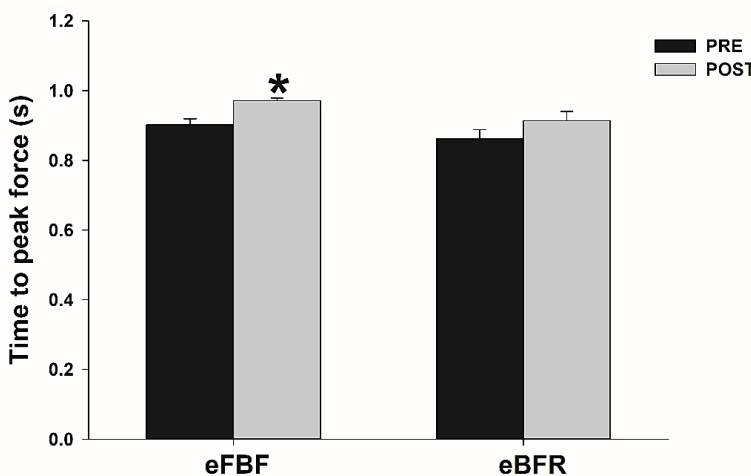

**Figure 2.** Mean ± SE of time to peak force before (PRE) and immediately after (POST) the isometric intermittent handgrip exercise until failure carried out with free blood flow (eFBF) or blood flow restriction (eBFR). (*) Significantly different from PRE at the same blood flow condition.

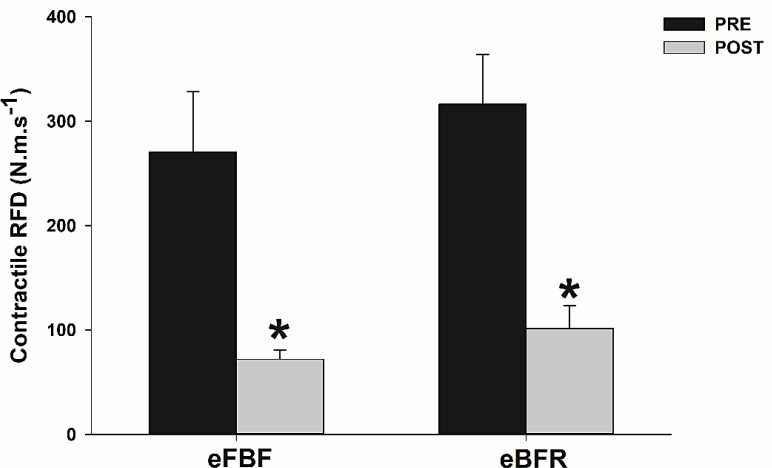

**Figure 3.** Mean ± SE of rate of force development (RFD) before (PRE) and immediately after (POST) the isometric intermittent handgrip exercise until failure carried out with free blood flow (eFBF) or blood flow restriction (eBFR). (*) Significantly different from PRE at the same blood flow condition.

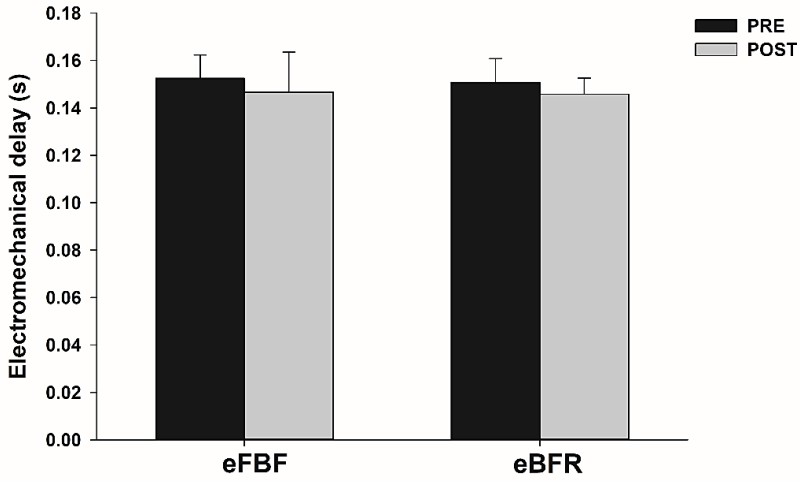

**Figure 4.** Mean ± SE of electromechanical delay (ED) before (PRE) and immediately after (POST) the isometric intermittent handgrip exercise until failure carried out with free blood flow (eFBF) or blood flow restriction (eBFR).

## 3. Discussion

The aim of the present study was to evaluate ED and TPF after an IIHE until failure, comparing the exercise protocol with and without BFR. Our results show a significant impairment in muscle performance (peak force and RFDpeak) and no difference between blood flow conditions (i.e., eBFR vs. eFBF) of the analyzed parameters. Furthermore, in both conditions, ED was not changed after exercise, while TPF increased significantly only after eFBF.

Muscle fatigue in humans can be defined as an acute, task-dependent reduction in the ability to generate voluntary force [28]. Some intracellular changes, such as impairments in sarcolemma action potential propagation, calcium output from the sarcoplasmic reticulum, and cross-bridge formation [21,29], comprise the local (i.e., muscular) fatigue mechanisms. Furthermore, changes in myotendinous unit compliance are observed after fatigue conditions, impairing the force transmission from the muscles to the bones [30]. Higher EMG activity is also observed during and immediately after fatiguing tasks, serving as a marker of a central (i.e., neural) mechanism. Interestingly, eBFR induced higher EMG activity after a bout of IIHE at the same submaximal load (45% of maximal voluntary isometric force) until failure than eFBF [12]. Thus, we hypothesized that eBFR could induce greater ED and TPF impairments than eFBF when exercise is carried out until failure. However, we did not observe significant difference between the studied conditions (i.e., BFR vs. FBF) in ED and TPF. Our findings are in agreement with a previous study that showed a similar level of fatigue markers (force coefficient of variation and amplitude of EMG signal) recorded during a submaximal (45% of MVIF) IIHE bout with BFR and FBF until failure [31]. Additionally, no significant difference was identified in the ED after an ankle dorsiflexion exercise performed at 60% MVIF until failure in hypoxia and normoxia [32]. Thus, together, the results indicate that blood flow condition does not influence the ED impairment after task failure.

The nature of the used exercise protocol to induce fatigue is the main factor that may explain the absence of significant changes in the ED, independent of blood flow condition. The IIHE included periods of work interspersed with brief moments of recovery, which allow the delivery of oxygen to the muscle for the subsequent work period [33]. In addition, blood supply may be enhanced by the "muscle pump" during recovery periods and by the action of chemical vasodilation mediators, balancing the offer/demand muscle oxygen supply during the exercise protocol [34,35].

TPF increased significantly only after exercise with eFBF, indicating that muscle contractile velocity tended to be more affected by fatigue in this condition. This finding can be explained by the longer time to failure during eFBF observed in the present and in other studies [36], which may have generated a slightly greater fatigue, with possible fatigue-induced impairment in $Ca^{2+}$ ion release by the sarcoplasmic reticulum and alteration in the coding rate of motor units [37,38]. It is worth noting that previous studies have not evaluated the TPF, thus making it difficult to compare our results with other studies analyzing the temporal characteristics of force development after eFBF was performed until failure.

## 4. Materials and Methods

### 4.1. Study Design and Ethical Aspects

This is a randomized crossover clinical trial. All procedures were approved by the local ethics committee (CAAE: 36832814.9.0000.5208) according to the Declaration of Helsinki. The purpose and procedures of the experiment were explained, and consent was obtained prior to the commencement of the experiment. Each subject underwent experimental procedures under the same instructions and conditions. This trial was prospectively registered (Clinical Trials N° NCT02384161). The CONSORT diagram of the study is presented in Figure 5.

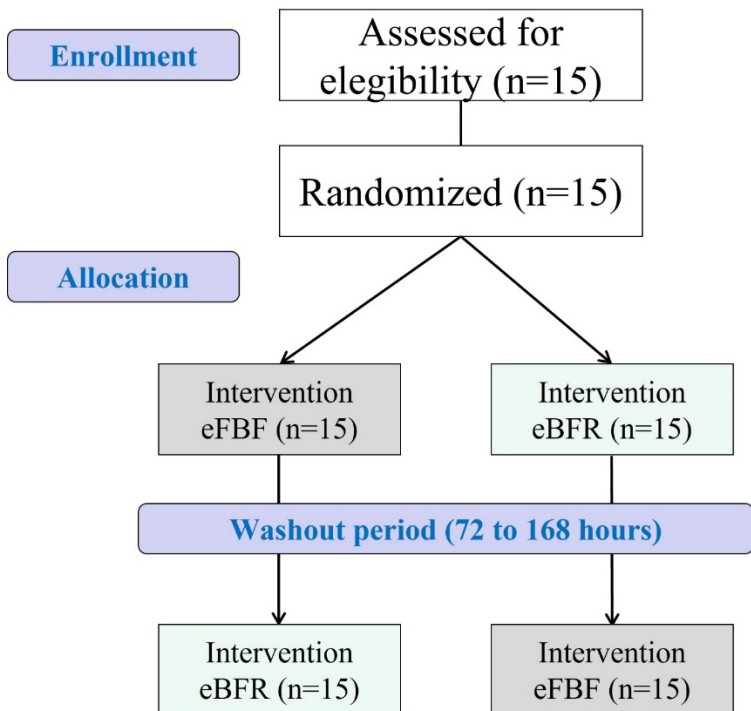

**Figure 5.** Adapted CONSORT diagram of the study. eFBF: exercise with free blood flow; eBFR: exercise with blood flow restriction.

### 4.2. Participants

Fifteen volunteers met the following criteria to be included in our study: male, aged between 18 and 40 years, and classified as active or very active according to the International Physical Activity Questionnaire (IPAQ) [39]. Volunteers with a body mass index (BMI) less than $18.5 \, \text{kg/m}^2$ and greater than $30 \, \text{kg/m}^2$, smokers, using vasoactive drugs, with a recent history of fracture in the upper limbs, or with cardiovascular or neurological disease were excluded, as well as any other condition that prevented them from performing the exercise.

### 4.3. Randomization, Allocation, and Blinding

Volunteers were submitted to two bouts of IIHE with different blood flow conditions. The interval between two successive bouts ranged from 72 h (minimum interval) and 1 week (maximum interval). A computer-generated random number table (http://www.randomization.com/ (accessed on 20 October 2015) was used to define the order of blood flow condition for each volunteer. An investigator who was not involved in the recruitment, intervention, or assessment of subjects was designed to generate the random allocation. Opaque sealed envelopes were used to conceal the allocation. To blind all procedures, the researcher responsible for maximum voluntary isometric force (MVIF) assessment and IIHE (researcher 1) did not know volunteer allocation. Evaluators determining (researcher 2) and applying/controlling restriction pressure (researcher 3) did not participate in the randomization. The volunteers were not informed about the applied blood flow condition and were instructed not to report the perception of pressure, keeping researcher 1 blinded.

### 4.4. Experimental Procedures

The volunteers were instructed to avoid strenuous exercise and ingestion of caffeine and alcohol 24 h prior to the data recording. Each subject participated in four laboratory sessions. During the first session, anthropometric data (age, height, body mass, and body mass index), physical activity level through IPAQ, and systemic arterial pressure were taken. Blood pressure was measured using a sphygmomanometer (Premium®, Duque de Caxias, RJ, Brazil) and a stethoscope (Diasyst®, Itu, SP, Brazil). The Edinburg Inventory [40]

was used in order to assess limb dominance. The determination of the total occlusion pressure (TOP) of the brachial artery at rest and familiarization with the evaluation of the MVIF and the IIHE protocol were carried out on the second day. On the third and fourth days, the interventions (eBFR or eFBF) were applied. The detailed description of the procedures is presented below.

### 4.4.1. Total Restriction Pressure Evaluation

The TOP was obtained using a Doppler ultrasound model SonoAcerR3 (Samsung Medison®, Seoul, Republic of Korea). Initially, the volunteers were positioned in dorsal decubitus with shoulder abduction (90°), complete elbow extension, and forearm supination, remaining at rest in an acclimatized and quiet room for ten minutes. Then, the transducer (linear transducer, 12 MHz; (Samsung Medison®, Seoul, Republic of Korea)) was positioned in the elbow flexor crease of the dominant arm, where visual and auditory signals indicated the presence of arterial pulse. A pressure cuff was inflated according to previous protocol [41]. The TOP was determined as the lowest pressure required to completely occlude arterial blood flow, and the partial occlusion pressure (POP) value was defined as 50% of the TOP. After TOP measurement, the volunteers were familiarized with the MIVF assessment and the IIHE protocol.

### 4.4.2. Maximum Voluntary Isometric Force (MVIF) and Surface Electromyography (sEMG)

Seventy-two hours after the familiarization session, the MIVF and the EMG activity were measured with the volunteers in the same position used to determine the TOP. The MIVF was evaluated using a hand dynamometer (*DM 100—Miotec*, Porto Alegre, RS, Brazil) with a 1000 N capacity (sampling frequency of 2.0 kHz). After resting for five minutes, they were asked to perform three maximal isometric voluntary contractions with 1 min rest between contractions. They were instructed to contract as fast and hard as possible, reaching maximum force between 1 and 3 s. All MIVFs were obtained with free blood flow. The highest value obtained between the three contractions was used as the reference value to obtain the target load (45% of MIVF) to be sustained during the IIHE. The cutoff point to define task failure was 30% of MIVF, when volunteers were unable to sustain the isometric efforts greater than 30% of MIVF, despite the verbal encouragement to achieve and sustain the target force (i.e., 45% of MIVF).

The EMG signal was recorded using a four-channel module (Miotool 400, Miotec Equipamentos, Biomédica, Porto Alegre, RS, Brazil), with total amplification gain of 1000 and common mode rejection rate of 110 dB. The analog signal was digitized by a 14-bit converter, whose recording has a sampling frequency of 2 KHz for each channel and an online bandpass filter of 3–500 Hz (Butterworth, fourth order). Pre-amplified (x100) bipolar superficial and circular adhesive electrodes of Ag/AgCl were used with interelectrode (center-to-center) distance of 20 mm. The skin was shaved and cleaned with alcohol, and anatomical landmarks (proximal third of the forearm, between the styloid processes of the radius and the humeral medial epicondyle) were determined to place electrodes over the flexor digitorum superficialis forearm and flexor carpi radialis, following the SENIAM [42] recommendations. A reference electrode was positioned in the caudal region of the sternal notch.

### 4.4.3. Intermittent Isometric Handgrip Exercise (IIHE)

After MVIF measurement, the volunteers performed the IIHE with or without blood flow restriction, respecting a wash-out period of at least 72 h and no longer than 1 week between the protocols [31]. The IIHE was performed only with the dominant limb, in the same position in which the TOP was obtained. The volunteers performed cycles of 10 s contractions (active cycle) interspersed by 5 s of rest (rest cycle). They were instructed to reach and maintain the target force (i.e., 45% of MVIF) during the active cycles, and relax (force = 0) during the rest cycles. The active and rest cycles were guided by timed beeps. The force–time curve recording was done with a digital hand dynamometer and projected

on the ceiling to give visual feedback to the volunteer along the IIHE protocol. Task failure was defined as the inability to maintain the target force above 30% of the MVIF in three consecutive active cycles. This cutoff point to declare task failure was identified by the evaluator and was not informed to the volunteers [27]. The time to failure during the IIHE with each blood flow condition was recorded for between-condition comparison.

The order of each protocol (i.e., eBFR vs. eFBF) was random for each volunteer, as described previously. When allocated in the eBFR condition, the pressure cuff was inflated immediately before and deflated immediately after the end of exercise protocol. The cuff pressure in the eFBF condition was defined as the pressure just enough to fit the cuff to the arm (~10 mmHg), maintaining free blood flow to the forearm. One minute after the end of exercise, the force and EMG activity were reevaluated.

### 4.5. Data Analysis (Time to Peak Force, Rate of Force Development, and Electromechanical Delay)

The peak force and the TPF were extracted from the force–time curve recorded during the MVIF. The strain gauge signal extracted from the force–time curves was smoothed by a digital fourth-order, zero-lag Butterworth filter, with a cutoff frequency of 15 Hz [27,43,44]. A specific algorithm identified the onset of the force–time curve, the peak force, and the elapsed time from the onset to the peak force (i.e., TPF). The onset of muscle contraction was defined as the time point at which the force curve exceeded the baseline by 2.5% of the difference between baseline force and the MVIF (i.e., maximum handgrip force) [27,43,44]. The RFDpeak was calculated by the peak force/time ratio from the contraction onset to peak force.

The EMG signal was smoothed by a digital fifth-order, zero-lag Butterworth filter, with a cutoff frequency of 10 Hz. The rectified EMG signal was used to identify the onset of EMG activity, which was defined as the time point at which the EMG signal exceeded 3x the SD of the baseline signal. The ED was defined as the difference between force–time curve onset and EMG activity onset.

Force–time curve and EMG signal analyses were conducted using specific routines developed in MATLAB 7.0.1® (MathWorks Inc., Portola Valley, CA, USA).

### 4.6. Statistical Analysis

The characteristics of volunteers were summarized using mean $\pm$ SD (standard deviation) (Table 1). Student's *t*-test was used to compare the time to failure during the IIHE with FBF and BFR. A two-way analysis of variance (two blood flow conditions $\times$ two measures) was used to compare peak force, TPF, RFDpeak, and ED obtained before (PRE) and immediately after the IIHE until failure carried out with FBF and BFR. Significant ANOVA results were followed by appropriate post hoc tests with Bonferroni corrections. A significance level of $p \leq 0.05$ was used to identify statistical significance. Data are graphically reported as means $\pm$ SE (standard error), and as mean difference with a respective 95% confidence interval in the text. All statistical procedures were carried out using SPSS v.21 (SPSS Inc., Chicago, IL, USA).

**Table 1.** Subject characteristics.

| Variable | Mean $\pm$ SD |
| --- | --- |
| Age (Years old) | 21.87 $\pm$ 2.13 |
| Height (m) | 1.76 $\pm$ 0.06 |
| Body weight (kg) | 77.07 $\pm$ 9.91 |
| BMI (kg/cm$^2$) | 24.67 $\pm$ 2.13 |
| SBP (mmHg) | 120.45 $\pm$ 11.27 |
| DBP (mmHg) | 76.13 $\pm$ 6.43 |
| TOP (mmHg) | 125.67 $\pm$ 10.33 |
| POP (mmHg) | 62.83 $\pm$ 5.16 |

BMI = Body mass index; SBP = systolic blood pressure; DBP = diastolic blood pressure; TOP = total occlusion pressure; POP = partial occlusion pressure.

## 5. Limitations

Some limitations could be listed: (1) small sample size, which may have influenced the high data variability; (2) sample composed only of young, healthy men, limiting extrapolation of our results to other populations; (3) the amount of blood flow restriction was determined from a single resting TOP, and it is possible that the amount of BFR decreased during exercise [45] and therefore minimized the magnitude of induced fatigue.

## 6. Conclusions

The TPF and RFDpeak were impaired following an IIHE protocol carried out until failure, while ED was not. Our data suggest the predominance of fatigue-related central (i.e., neural) factors in the applied exercise protocol. The absence of difference between blood flow conditions in all studied parameters suggests a similar post-exercise fatigue level. Our results expand the knowledge about the neuromuscular mechanisms of exercise-induced muscle fatigue when it is conducted with BFR, and future studies should investigate whether more intense BFR exercise protocols (continuous isometry or use of higher occlusion pressures) could induce a greater impact on force–time curve parameters and ED.

**Author Contributions:** Conceptualization, M.S.C. and R.P.; methodology, M.S.C., M.L., R.S.d.S. and R.P.; formal analysis, M.S.C., M.L., R.S.d.S., M.M. and R.P.; investigation, M.S.C. and R.P.; data curation, M.S.C., M.L. and R.S.d.S.; writing—original draft preparation, M.S.C., M.L., R.S.d.S. and R.P.; writing—review and editing, M.L., R.S.d.S., M.M. and R.P.; visualization, M.S.C., M.L., R.S.d.S. and R.P.; supervision, M.M. and R.P.; project administration, M.S.C. and R.P. All authors have read and agreed to the published version of the manuscript.

**Funding:** This research received no funding.

**Institutional Review Board Statement:** The study was conducted in accordance with the Declaration of Helsinki and approved by the local Ethics Committee (protocol code CAAE: 36832814.9.0000.5208).

**Informed Consent Statement:** Informed consent was obtained from all subjects involved in the study.

**Data Availability Statement:** The data presented in this study are available on request from the corresponding author.

**Acknowledgments:** The first author would like to thank the National Council for Scientific and Technological Development (CNPq) for the postdoctoral fellowship (Junior Post-Doctorate—PDJ 25/2021; Process: 150483/2022-8).

**Conflicts of Interest:** The authors declare no conflict of interest.

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
