# Peer review of "Effects of Blood Flow Restricted Exercise on Electromechanical Delay and Time to Peak Force after Task Failure: A Randomized Crossover Trial"

_muscles, doi:10.3390/muscles1030014_

Round 1

Reviewer 1 Report

The present study aimed to compare the electromechanical delay and time to peak force after an intermittent isometric task until failure with blood flow restriction. The study found that while time to peak force was significantly increased, there was no significant change in electromechanical delay, suggesting that other mechanisms was responsible for inducing fatigue during BFR training. The results of this study allow for better understanding about BFR, which would be of interest to practitioners using this training method. The study is generally well written with appropriate methodology for the objective of the study. I support the publication of the manuscript in its current form.

Author Response

Dear reviewer, thank you for the positive comments.

Reviewer 2 Report

Dear author’s

I was pleased to review your manuscriptand the subject is intereasting.

The study sample is very small but the research may continue.

Explain the novelty of your study.

Author Response

Dear reviewer, thank you for the time and effort you put forth in reviewing our submission. We have revised our manuscript according to your suggestion to “explain the novelty of our study”, and we feel that the paper is now much stronger as a result of your input.
Additionally, to facilitate the editing revisions all changes done in the manuscript are highlighted (see our conclusion section). We hope that our corrections and justifications meet your approval, but, if necessary, we are willing to make further corrections if need be.

Reviewer 3 Report

This study investigated the electromechanical delay and the time to peak force after intermittent hand-grip exercise in two conditions: with and without blood flow restriction. The study is well written, and the proper methodology was used. The Figures used are appropriate and clear. The conclusions seem logical and not overstretched.

A few  minor comments follow.

Was there any correlation between the initial muscle strength or the time to failure, on the one hand, and the % change in TPF or the RFD on the other? Please provide this analysis.

Introduction. Line 37. ....>50%.... Perhaps the authors mean < instead of >.

Figure. The legends of the horizontal axis should be eFBF and eBFR since these are used in the text.

Reference 31. Information is missing.

Author Response

Dear reviewer,
Thank you for the time and effort you put forth in reviewing our submission. We have revised our manuscript according to your concerns and suggestions, and we feel that the paper is now much stronger as a result of your input.
We have answered the questions as you will find below. Additionally, to facilitate the editing revisions all changes done in the manuscript are highlighted. We hope that our corrections and justifications meet your approval, but, if necessary, we are willing to make further corrections if need be.

This study investigated the electromechanical delay and the time to peak force after intermittent hand-grip exercise in two conditions: with and without blood flow restriction. The study is well written, and the proper methodology was used. The Figures used are appropriate and clear. The conclusions seem logical and not overstretched.
Response: Thanks for the positive comments.

A few  minor comments follow.

Was there any correlation between the initial muscle strength or the time to failure, on the one hand, and the % change in TPF or the RFD on the other? Please provide this analysis.
Response: Thank you for the relevant comment. The muscle strength and time to failure data were not normalized (i.e., corrected) by initial values (i.e., measures obtained PRE-intervention) since we used a crossover design where the same subjects were submitted to both interventions and there were no differences between PRE-interventions measures. Additionally, when we opt to compare absolute values, instead of normalized values, we can report the mean difference with its respective 95% confidence intervals, allowing the reader to infer the magnitude of the difference between interventions.

Introduction. Line 37. ....>50%.... Perhaps the authors mean < instead of >.
Response: Yes, thank you for reporting this mistake. It was corrected now.

Figure. The legends of the horizontal axis should be eFBF and eBFR since these are used in the text.
Response: It was done as requested.

Reference 31. Information is missing.
Response: Data was added.